# An Optimal Strain Gauge Layout Design for the Measurement of Truss Structures

**DOI:** 10.3390/s23052738

**Published:** 2023-03-02

**Authors:** JungHyun Kyung, Hee-Chang Eun

**Affiliations:** Department of Architectural Engineering, Kangwon National University, Chuncheon 24341, Republic of Korea

**Keywords:** sensor measurement, Fisher information matrix, strain gauge, optimal sensor layout, effective independence, Guyan condensation method

## Abstract

Sensor measurements diagnose and evaluate the structural health state. A sensor configuration with a limited number of sensors must be designed to monitor sufficient information about the structural health state. The diagnosis of a truss structure composed of axial members can begin with a measurement by the strain gauges attached to the truss members or by the accelerometers and displacement sensors at the nodes. This study considered the layout design of the displacement sensors at the nodes for the truss structure by using the effective independence (EI) method based on the mode shapes. The validity of the optimal sensor placement (OSP) methods depending on their synthesis with the Guyan method was investigated by the mode shape’s data expansion. The Guyan reduction technique rarely affected the final sensor design. A modified EI algorithm based on the strain mode shape of the truss members was presented. A numerical example was analyzed, showing that the sensor placements were affected depending on the displacement sensors and strain gauges. Numerical examples illustrated that the strain-based EI method without the Guyan reduction method has the advantage of reducing the number of sensors and providing more data related with the displacements at the nodes. The measurement sensor should be selected when considering structural behavior, as it is crucial.

## 1. Introduction

With the recent advent of various types of innovative sensors, many measurement methods for diagnosing structural health state have been developed and applied in actual structures [1]. Structural health monitoring (SHM) aims to improve structural durability and diagnose the structural state from the point of view of maintenance. The measured data are significantly informative for structural damage detection, system identification, and data expansion.

Many types of sensors such as accelerometers, displacement sensors, and strain gauges may be considered for optimal monitoring. Increasing the number of measurement sensors for SHM can identify more accurate structural performance. However, measurement by a large number of sensors is impractical and uneconomical for collecting field data. A desirable monitoring technique would be to evaluate structural performance using a minimized number of sensors and their layout. To collect sufficient information, a sensor layout design should include the pertinent selection of sensors for measurement and the number and locations of the sensors.

The truss structure consists of the axial members with an axial-load-carrying capacity. There have been many attempts related to optimal measurement sensors in truss structures [2,3,4,5,6,7]. The axial strain of a truss member can be formulated by the displacements at both ends. Its performance evaluation being based on axial strains rather than nodal displacements would be more realistic, because then the sensors could collect more information about the whole structure.

Many OSP algorithms such as the Fisher information matrix (FIM), modal assurance criterion, and singular value decomposition have been presented [8,9,10,11,12,13,14,15]. Starting from the lowest number of target modes, OSP techniques track the sensor locations corresponding to a limited number of sensors. 

The majority of optimal sensor design techniques utilize mode shape data that are treated as structural characteristics. The EI method proposed by Kammer [16] was developed from the independence of the mode shapes and has been widely used in OSP techniques. The EI approach is an iterative method for designing the best target sensor layout. After excluding the row of the mode shape matrix corresponding to the diagonal element of the EI matrix with the smallest value at each iteration, the next iteration is performed. The EI iterative technique ultimately selects the positions to maximize both the spatial independence and signal strength of the targeted mode shapes. This technique is carried out by maximizing the determinant of the FIM [17,18,19,20,21,22]. The sensors should be located at the common nodes of the high EI index. Friswell and Castro-Triguero [23] investigated the linear independence of the modes of the mode-shape-based EI method. Using four sensor placement methodologies and considering uncertain parameters, Castro-Triguero et al. [24] studied the sensor configuration. 

Most OSP algorithms regard the first few modes as target modes. The row of the mode shape matrix at each iteration can be excluded or retained in the EI method. The Guyan condensation algorithm has been utilized for designing optimal measurement locations. If the rows of the mode shape matrix removed at each iteration do not affect the next iteration, the remaining substructure exhibits the vibration mode of a rigid body. Then, the EI index at the final iteration approaches 1.0 because the effect of stiffness gradually decreases. If the removed rows can be considered in the next iteration, more accurate results can be predicted because they are distributed between the other degrees of freedom (DOFs). 

Based on the genetic algorithm, Guyan reduction approach, and structural subsection technique, Lu et al. [25] proposed an OSP method to capture complete and accurate modal information. Kammer and Peck [26] investigated an appropriate mass-weighting approach using an iterative Guyan expansion and the EI sensor set expansion. Yang and Xia [27] proposed a multi-objective optimization method to select the sensor positions based on the EI method and dynamic condensation approach. Jaya et al. [28] provided an OSP algorithm for system identification to reduce the correlation between the different modes. Liu et al. [29] established their sensor placement by formulating an optimization problem by the trace of the inverse of the Bayesian FIM. Chen et al. [12] developed a hybrid OSP method by applying the optimization principle of the modal assurance criterion matrices and EI vectors. Jiang et al. [30] mentioned the effect of different weighting coefficients on the maximization of the FIM and the physical significance of the re-orthogonalization of the mode shapes through QR decomposition in the EI formula. Blachowski et al. [15] provided an OSP method based on the objective function of the FIM matrix and the concept of structural topology optimization. He et al. [31] introduced the concept of the generalized equivalent stiffness, defined the importance coefficient of the component, and determined the sensor placement scheme according to statistical data. 

The strain gauge, which is one of the most widely used sensors, has been utilized as a measurement sensor for performing subsequent analysis such as damage detection and system identification to evaluate the health state of structures. Xiao et al. [32] identified the minimum number of sensors for static strain sensor placement and the optimal sensor layout based on an assumed set of applied static forces and the measured strains. Song et al. [33] constructed an OSP algorithm using reduced strain sensors to the satisfaction of the constraint conditions of the optimal sensor placement in a truss structure. Evaluating different sensor placement methods with different constraint variants by machine learning techniques, Rucebskis et al. [34] exhibited the possibility of detecting damage with a limited number of strain sensors and mode shapes. Zhao et al. [2] constructed an OSP model of strain sensors based on eigenvalue analysis through a model solution created using the Particle Swarm Optimization algorithm. Yi et al. [35] provided a hybrid algorithm using QR factorization, modal assurance criterion, and generalized genetic algorithm. Papadimitriou [36] presented an asymptotic approximation algorithm using lower and upper bounds for the optimal sensor configuration. 

The EI technique removes the DOFs to minimize both the spatial independence and signal strength of the targeted mode shapes from the candidate positions of the analytical model. By repeating this operation, the final sensor positions can be established. The effect of the DOFs removed from the master DOFs can be neglected in the next iteration or can be retained by the Guyan condensation method. This study compared the OSP layouts depending on the effect of the removed DOFs in a numerical example to design the OSP layout of a truss structure. The removed mode shapes had an insignificant effect on the next iteration when taking the Guyan reduction approach in a numerical example to handle the mode shape’s data expansion. 

The axial strain of a truss member is expressed as a function of the displacement components at both ends. Using strain gauges to investigate the structural health state of the truss structures, this study proposed the strain-based EI method based on strain measurement. A numerical example illustrated that the axial strain can be estimated by using more nodal displacement data and can reduce the number of sensors compared to the use of displacement sensors. The comparison of the OSPs depending on the displacement measurement sensors and strain gauge sensors was analyzed, which showed that the sensor placements were affected depending on the characteristics of the selected sensors. 

## 2. Formulation

### 2.1. Displacement Mode Shape-Based Approach

The structural health state can be determined by evaluating and analyzing the measured response data. The locations of measurement sensors should be designed to collect enough information about the structural performance. Mode shapes have been widely used as the measurement data for SHM. Measurement sensors are selected by considering the mechanical behavior of the structures such as bending, axial load, and twisting. 

The truss structure is composed of axial members to carry axial loads and to deform in the direction of the member axis. The mechanical behavior of the truss is described by the displacements at the joints or the axial deformation in the longitudinal direction. A measurement using the strain gauges can estimate more multiple-displacement information and is more economical than the displacement sensors.

Most OSP algorithms are derived from the structural dynamic vibration. The undamped dynamic equation of motion for the truss structure modeled by *n* DOFs can be written as
(1)$$M\ddot{u}+Ku=0,$$
where M and K represent the n×n mass and the stiffness matrices, respectively. u and $\ddot{u}$ denote the n×1 displacement and the acceleration vectors, respectively. Transforming a displacement vector to a modal displacement vector via the mode shape matrix can be written as
(2)u=ϕy,
where y is the m×1 modal coordinate vector, and ϕ represents the $n\times m\;\left(n>m\right)$ mode shape matrix. m is the lowest number of mode shapes, and n is the number of the entire DOFs at all the nodes of the truss. 

The EI method was developed based on the mode shape matrix being related to the dynamic characteristics. The optimal sensor was designed using a lower bound on the variance of an unbiased estimator. The covariance matrix for the difference between the observed and estimated displacement vectors, **P**, can be expressed by
(3)$$P=E\left[\left(u-\hat{u}\right){\left(u-\hat{u}\right)}^{T}\right]={\left[{\left(\frac{\partial \mu }{\partial u}\right)}^{T}{\left[\psi_{0}^{2}\right]}^{-1}\left(\frac{\partial \mu }{\partial u}\right)\right]}^{-1}=F^{-1},$$
where E is the expected value, $\hat{u}$ denotes an unbiased estimator of the vector u, and μ is a measurement column vector. $\psi_{0}^{2}$ represents the stationary Gaussian white variance matrix, and F is the FIM to be expressed as the product of the mode shape matrix corresponding to the master DOFs and its transpose: $F=\phi^{T}\phi$. The FIM is symmetric, positive semidefinite, and full rank. 

The EI approach is an iterative method to track the final target sensor positions from their candidate locations by optimizing the linear independence of the mode shapes. This approach requires the iterative computation of the EI index, expressed by the target mode shape matrix, and its reduced form. The EI index is calculated by the diagonal elements of the orthogonal project matrix as follows:(4)$$E=\phi F^{-1}\phi^{T}.$$

The diagonal element in matrix **E** represents the fraction contribution of the sensor locations and the linear independence of the modes. The measurement sensors are selected depending on the degree of the fraction contribution of the sensor locations and the linear independence of the modes. If the value of the diagonal element of matrix **E** corresponding to the master DOFs is relatively small, it is excluded from the candidate sensor locations. The same process is repeated by deleting the DOF of the low value in the diagonal elements of matrix **E** from the candidate sensor DOFs. The final sensors are positioned at the DOFs with a relatively high EI index of the diagonal elements. 

The rows in matrix **E** corresponding to the DOFs removed from the candidate sensor positions are neglected in the next iteration. As the candidate sensor positions gradually approach the final sensor layout, the effect of the mode shape matrix corresponding to the slave DOFs insignificantly increases when designing the sensor locations. Thus, it is necessary to investigate whether the incorporation of the removed mode shape matrix by the Guyan condensation method affects the final OSP. The effectiveness of the G-EI technique to synthesize the EI technique and the Guyan reduction method was examined. 

Inserting Equation (2) into Equation (1) by taking the eigenvalue analysis on the resulting equation, dividing the entire DOFs into the retained master and the removed slave DOFs, and arranging the result can be written as
(5)$$\left[\begin{array}{cc} K_{mm} & K_{ms}\\ K_{sm} & K_{ss} \end{array}\right]\left[\begin{array}{c} \phi_{m}\\ \phi_{s} \end{array}\right]-\omega^{2}\left[\begin{array}{cc} M_{mm} & 0\\ 0 & M_{ss} \end{array}\right]\left[\begin{array}{c} \phi_{m}\\ \phi_{s} \end{array}\right]=\left[\begin{array}{c} 0\\ 0 \end{array}\right],$$
where the subscripts *m* and *s* denote the master DOF and slave DOF, respectively. ω denotes the circular natural frequency. Assuming the mass matrix corresponding to the slave DOFs is $M_{ss}\approx 0$, the second equation of Equation (5) can be simplified as
(6)$$\phi_{s}=-K_{ss}^{-1}K_{sm}\phi_{m}.$$

The mode shape matrix of the entire DOFs is formulated by the master DOFs’ mode as
(7)$$\phi =\left[\begin{array}{c} \phi_{m}\\ \phi_{s} \end{array}\right]=\left[\begin{array}{c} I\\ -K_{ss}^{-1}K_{sm} \end{array}\right]\phi_{m}.$$

In the second part of Equation (7), the slave DOFs’ mode shape matrix is described by the product of the stiffness matrix corresponding to the slave DOFs and the master DOFs’ mode shape matrix. The DOF with a low EI index is removed from the master DOFs, but the corresponding stiffnesses are utilized as the variables in the next iteration, as shown in Equation (7).

Substituting Equation (7) into Equation (4), the G-EI matrix corresponding to the master DOFs is simplified based on the Guyan condensation method. The effect of the slave DOFs’ stiffness matrix is included in the next iteration for the G-EI approach. Thus, the G-EI matrix is composed of only the mode shape matrix of the master DOFs. In the subsequent process, the DOF corresponding to the lowest value of the diagonal elements of the G-EI matrix is moved to the slave DOF. The final OSP is determined by repeating the same process. In the following example, the optimal sensor layouts obtained by using the EI approach and the G-EI approach are determined and compared. 

Example (1)

This example compared the OSP designs for the displacement sensors of the truss structure shown in Figure 1 using the EI and G-EI approaches. The member and node numbers are shown in Figure 1. The truss consisted of 12 nodes and 21 members and was simply supported. Each node had two displacement DOFs in the horizontal and vertical directions. The structure consisted of 21 displacement DOFs, excluding the boundary conditions at both ends. Each chord and diagonal member had lengths of 4 and 5 m, respectively, and the vertical member was 3 m. Each member had a modulus of elasticity of 200 GPa and a cross-sectional area of $5500{\;\mathrm{mm}}^{2}$. The first six mode shapes were utilized as target modes, and the same number of sensors and modes was assumed in the final sensor configuration. The entire DOFs were initially established as the sensor candidate positions. At each iteration, one master DOF was moved to the slave DOF, and the iteration was repeated until the final sensor number was reached for the master DOFs. 

Figure 2 compares the transition of the DOFs moving from the master DOFs to the slave DOFs at each iteration, as estimated by both methods. Figure 2 exhibits the same process until the ninth iteration. After the 10th iteration, both methods exhibited a slight difference in their transition due to the cumulative effect of the slave modes with the increase in the number of iterations. 

The final displacement sensors located at the DOFs are shown in Table 1 and Figure 3. The locations of the final displacement sensors based on the EI method were designed to be three each in the x and y directions. The sensors were equally distributed in each direction. The G-EI method requires five sensors in the y direction and one sensor in the x direction. The sensors were distributed by being concentrated in the y direction. 

The sensor positions obtained were investigated by the two methods, and they do not exactly match, but the sensors were similarly designed, except for in two locations. A slight difference occurred depending on whether or not the Guyan technique was applied. 

In both techniques, the displacement sensors in the x direction were removed or moved from the master DOFs during the first six iterations. After the seventh iteration, the displacement sensors were removed evenly in the x and y directions. The measurement data in the x direction did not significantly affect the subsequent analysis of the truss structure, such as data expansion, system identification, etc. 

It is necessary to verify the sensor design created by both methods. The superiority of both methods was compared in the data expansion of the mode shapes. The measured displacement modes at the OSP DOFs were expanded to the slave modes using the second part of Equation (7). Figure 4 displays the difference between the actual and estimated 15×6 slave mode shape matrices. In the plot, the errors deviated from the actual slave mode shape matrix occur regardless of whether the Guyan reduction method was applied or not. Both methods did not accurately estimate the slave mode shape matrix by data expansion when only using the displacement data measured in six sensor locations. The two methods showed similar maximum magnitude errors. 

Figure 5 and Figure 6 compare the mode shapes predicted by the EI and G-EI methods with actual mode shapes at unmeasured DOFs, respectively. It was shown in these plots that the mode shapes estimated by the EI and G-EI methods greatly deviated from the actual values except for a few DOFs. This discrepancy was due to the measurements at a few DOFs and the result of considering only six mode shapes that are fewer than the entire DOFs. It was determined in both methods that the inconsistency increased as the order of the mode increased. Therefore, it was evaluated that it could be improved by increasing the number of sensor measurements. 

The errors gradually reduce with an increase in the number of measurement positions. This can reduce the effect of the stiffness matrix and mode shapes belonging to the slave DOFs. Rather than increasing the number of displacement sensors, an optimal design using strain gauges related to multiple displacement DOFs was considered. In addition, the characteristics of the strain gauge were incorporated into the OSP algorithm. 

### 2.2. Strain Mode Shape-Based Approach

The structural performance of a truss member may be diagnosed by measuring the longitudinal strain instead of measuring the displacements at the nodes. Considering the axial deformation of a truss member, four displacement measurements at both nodes can be described by only one strain, as shown in Figure 7. The number of candidate sensor locations for the displacement measurement in a truss structure can be reduced by the strain gauges. In another sense, the strains are related to having more nodal displacement data than the number of strain gauges. The number of candidate positions for the strain gauges corresponds with the number of truss members. 

The strain sensor design for the truss structure began with the relationship between the displacements in the horizontal and vertical directions at nodes *i* and *j* and the strain in the axial direction. This is expressed by
(8)$$\varepsilon_{ij}=\frac{1}{L_{ij}}\left[\begin{array}{cc} \begin{array}{cc} -cos\theta & -sin\theta \end{array} & \begin{array}{cc} cos\theta & sin\theta \end{array} \end{array}\right]\left[\begin{array}{c} \begin{array}{c} u_{i}\\ v_{i} \end{array}\\ \begin{array}{c} u_{j}\\ v_{j} \end{array} \end{array}\right],$$
where L is the length of the truss member; u and v denote the displacement components in the *x* and *y* directions, respectively; and θ is the directional angle, which is counterclockwise relative to the horizontal axis. According to the relationships in Equation (8), the strain of the truss member can be estimated by the displacement at both ends. The axial strain can be estimated from the displacements at the nodes, but the displacements cannot be estimated from the axial strain. This indicates that the measurement positions for the strain gauges can be reduced.

The displacement vector in Equation (2) is transformed into the strain vector by using the relationship between the strain and displacements in Equation (8). It follows that
(9)ε=Ru=Rϕy,
where ε is the strain vector, and R is the coefficient rectangular matrix making the relationship between the nodal displacement vector and the axial strain vector. 

Modifying Equation (3) with the help of Equation (9), the strain-based covariance matrix can be formulated by
(10)$$P_{\varepsilon }=E\left[\left(\varepsilon -\hat{\varepsilon }\right){\left(\varepsilon -\hat{\varepsilon }\right)}^{T}\right]=F_{\varepsilon }^{-1},$$
where $\hat{\varepsilon }$ represents an unbiased estimator of the vector ε, and $F_{\varepsilon }$ is the FIM to be expressed by $F_{\varepsilon }={\left(R\phi \right)}^{T}R\phi$. The strain-based orthogonal project matrix can be written as
(11)$$E_{\varepsilon }=R\phi F_{\varepsilon }^{-1}{\left(R\phi \right)}^{T}.$$

The strain-based EI matrix consists of the member’s axial strain. In the same way as in the previous displacement-based EI method, the members were separated into the master and slave axial strain DOFs. Among the diagonal elements of the EI matrix of Equation (11), the row of the lowest EI index was removed, and the next iteration was calculated. The strain-based EI calculation did not require the Guyan reduction method because many displacement components were already reduced in the process of strain calculation, and the removed slave displacement modes did not have significant effect. Thus, by repeating the process, the final sensor design was obtained. 

Example (2) 

As in the previous example, this example estimated the optimal locations for the strain gauges in the truss structure of Figure 1. The nodal displacements were converted into the axial strains of the truss members using Equation (8). The strain-based sensor locations were estimated using the EI approach.

Starting with all truss members as sensor candidates, Figure 8 exhibits the members removed from the master members in iteration order. Figure 8 shows that the vertical members were removed in the first few iterations because they do not carry any forces and do not deform in the member axis. In fact, if the load does not directly act on the vertical member, it does not carry any load. The final strain gauges were designed to be located at members ①, ⑤, ⑨, ⑬, ⑯, and ㉑, as shown in Figure 9. The nodal displacements belonging to these members numbered 18 in total, and almost all the nodal displacements of the truss can be related to the six strain gauges. This indicates that the strain gauge measurement can collect more information.

The vertical displacement components at nodes 4, 9 and 11 selected as the OSP in the previous example were not related to the strain gauge sensor design. Since the vertical members corresponding to the vertical displacement components at nodes 4, 9 and 11 are zero-force members, they were not suitable for the strain gauge measurement locations. However, in the displacement-based approach, the missed displacement components at nodes 4, 9 and 11 affected the displacement modes of the other nodes. This phenomenon was due to the selected sensors and the sensor placements originating from the difference in the characteristics of both measurement sensors when considering the deformed shape of the structure. This indicates that the selection of the measurement sensor is crucial when measuring, considering that the deformed shape governs the structural behavior.

The displacement modes at the slave DOFs could be estimated by Guyan condensation method with measured displacement modes. However, in measurement by strain gauges, the displacement modes must be expanded using the measured strains as constraints. The expansion method was derived in the Appendix A. The displacement modes at the entire DOFs were estimated from the measured stains in the six optimal sensor members.

Figure 10 compares the displacement modes expanded using the data expansion method with the actual modes. It was shown that the expanded data included the errors deviated from the actual mode data. The inconsistency between the expanded displacement modes obtained from the strain-based EI approach and the actual modes was observed. The expansion method in the Appendix A does not estimate the accurate displacement modes. The constraints were the relationship between the six strain modes and the corresponding four displacement modes. The relationship between the strain and the displacement mode was not a one-to-one correspondence, but represented a relative relationship between them. It was analyzed that the discrepancy occurred in relative relationship rather than absolute relationship. 

Compared to the displacement modes obtained by the EI and G-EI methods in Figure 5 and Figure 6, it was observed that the displacement modes obtained from the strain-based approach were determined to be closer to the actual modes than the other methods. The same displacement modes were observed at several DOFs and large errors at only several DOFs. It was concluded that the strain gauge measurement can obtain closer results in relation to the measurement data in multiple DOFs than the displacement mode measurement sensor.

## 3. Conclusions

This study considered the OSP design for the truss structure by using the EI method based on displacement and strain mode shapes. The Guyan reduction technique rarely affected the final sensor placements despite requiring a complicated process. The OSPs estimated by the EI and G-EI methods did not exactly match. The validity of both methods was investigated by the mode shape’s data expansion at the slave DOFs. Both methods did not accurately lead to the actual mode shape at the slave DOFs. Displacement measurements for more DOFs by more sensors were required. 

The OSP by the strain gauge was considered as an alternative for obtaining more displacement measurements. An example compared the sensor layouts by the displacement sensors at the nodes and the strain gauges on the truss members. Inconsistent results were detected between both methods. The strain-based EI method has the advantage of reducing the number of sensors and providing more data. Moreover, it was observed that the sensor locations depended on the selection of sensors and the characteristics of the selected sensors. 

## Figures and Tables

**Figure 1 sensors-23-02738-f001:**
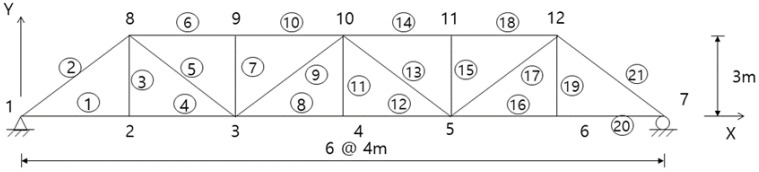
A simply supported truss structure for a numerical example. The circle numbers indicate the member number.

**Figure 2 sensors-23-02738-f002:**
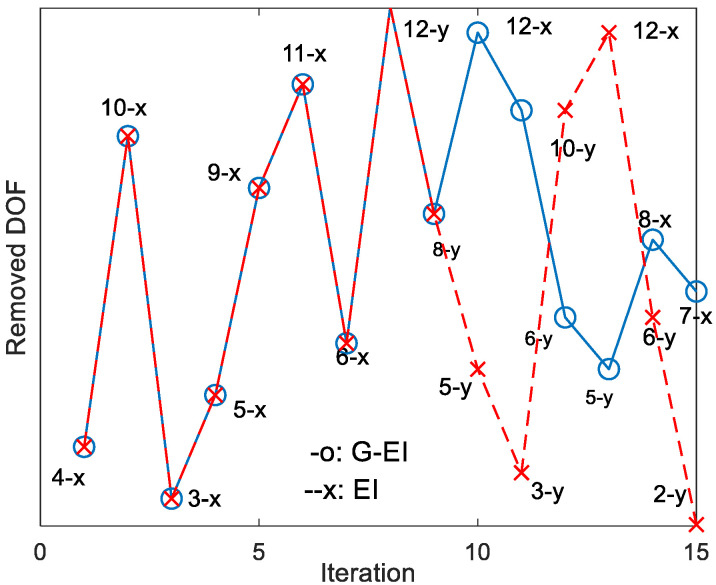
Transition of removed DOFs according to the iteration. The number indicates the node number, and the letters x and y indicate the measurement direction. The blue solid line indicates the G-EI and the red dotted line the GI.

**Figure 3 sensors-23-02738-f003:**
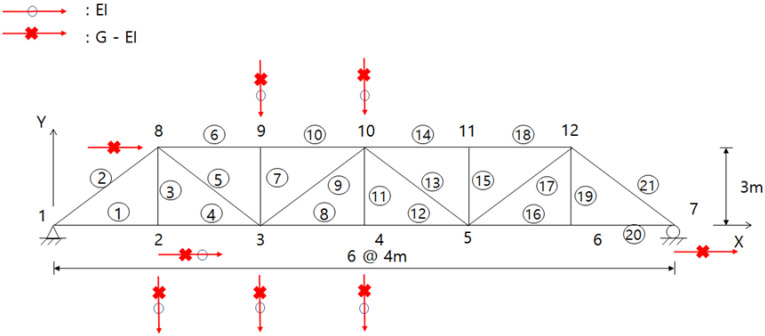
Final displacement sensor locations using both methods.

**Figure 4 sensors-23-02738-f004:**
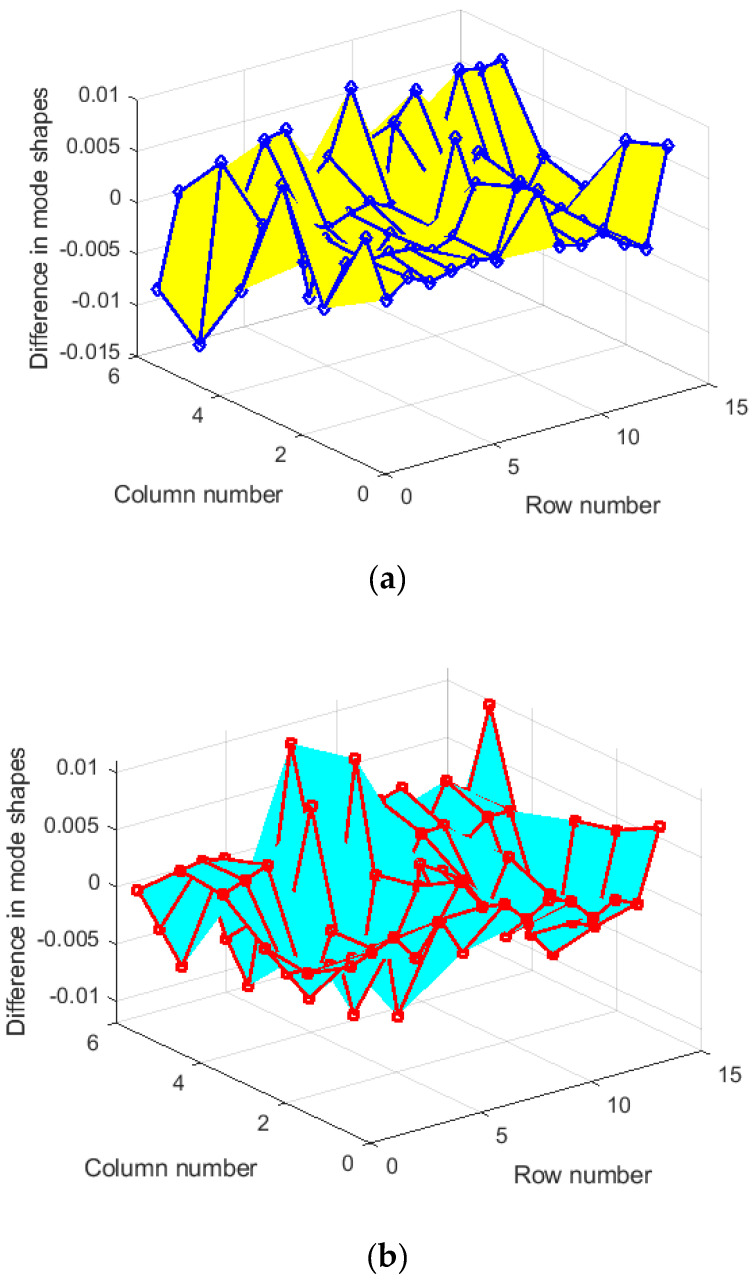
Difference between actual and estimated slave mode shape matrix. (**a**) EI method; (**b**) G-EI method. The values in the z axis indicate its corresponding row and column in the mode shape matrix.

**Figure 5 sensors-23-02738-f005:**
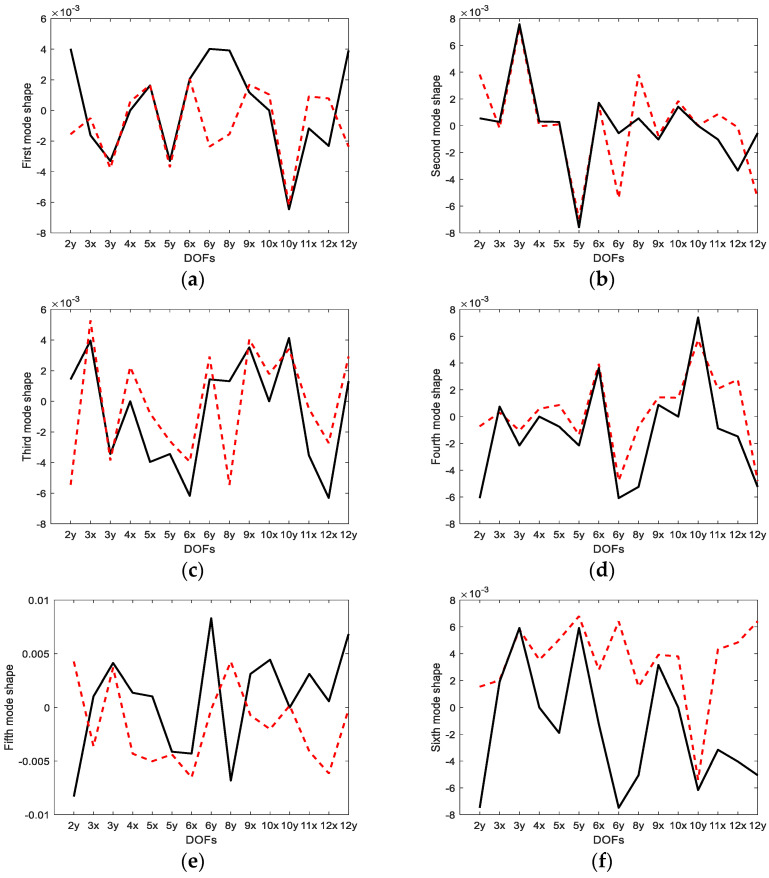
Comparison of estimated and actual mode shapes (EI approach) at slave DOFs: (**a**) first mode; (**b**) second mode; (**c**) third mode; (**d**) fourth mode; (**e**) fifth mode; (**f**) sixth mode. The solid line indicates the actual mode shape and the dotted line is the estimated mode shape.

**Figure 6 sensors-23-02738-f006:**
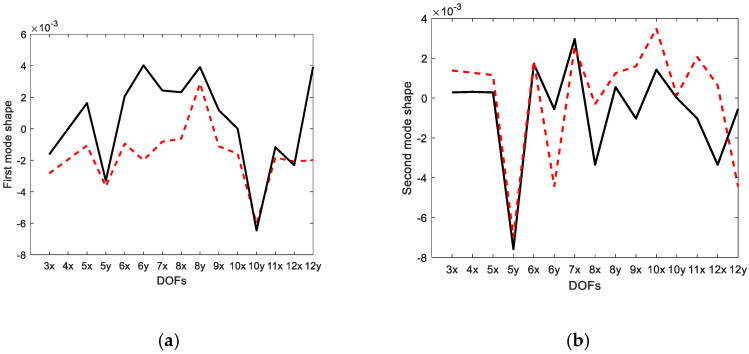
Comparison of estimated and actual mode shapes (G-EI approach) at slave DOFs: (**a**) first mode; (**b**) second mode; (**c**) third mode; (**d**) fourth mode; (**e**) fifth mode; (**f**) sixth mode. The solid line indicates the actual mode shape and the dotted line is the estimated mode shape.

**Figure 7 sensors-23-02738-f007:**
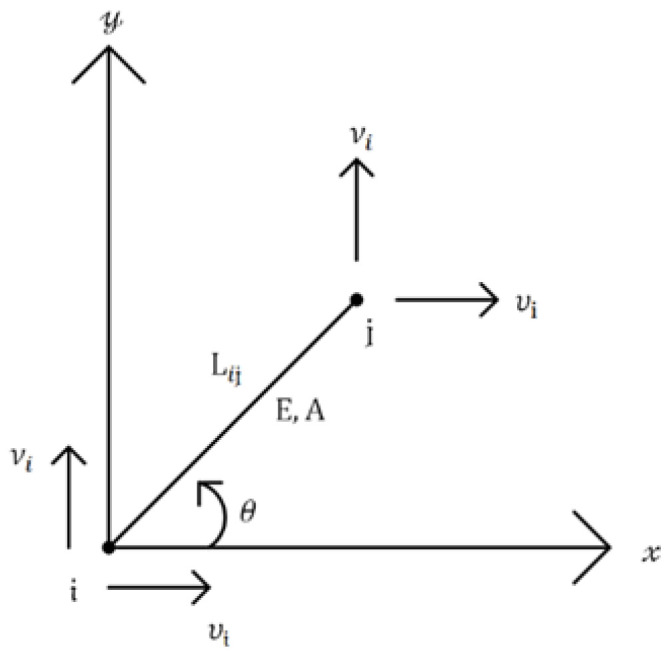
Displacement DOFs in the horizontal and vertical directions.

**Figure 8 sensors-23-02738-f008:**
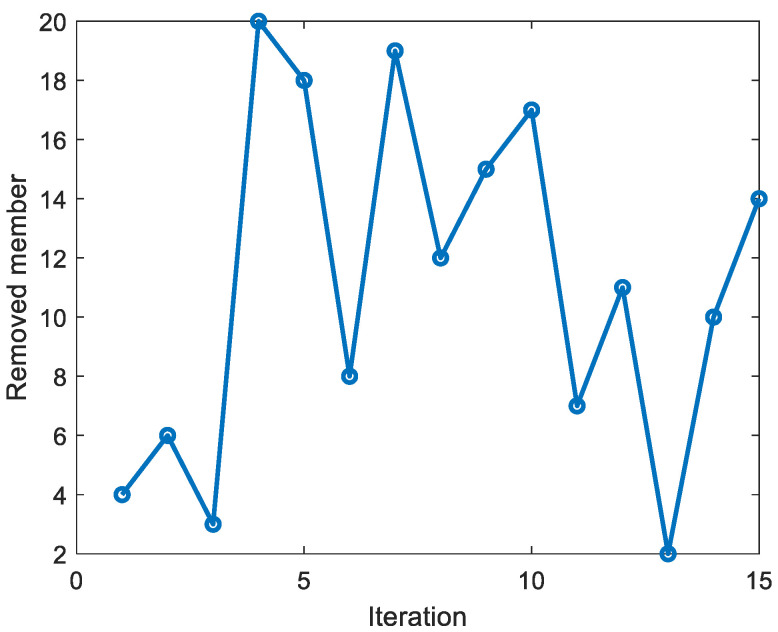
Transition of removed members according to the iteration.

**Figure 9 sensors-23-02738-f009:**
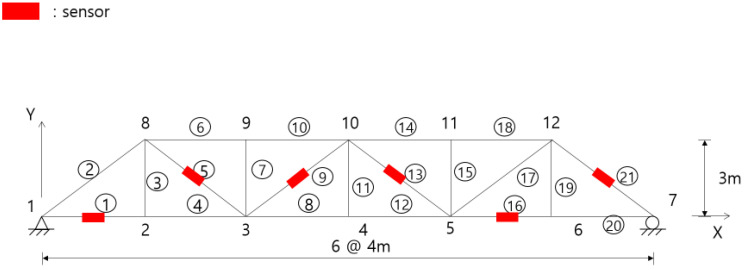
Final strain sensor layout.

**Figure 10 sensors-23-02738-f010:**
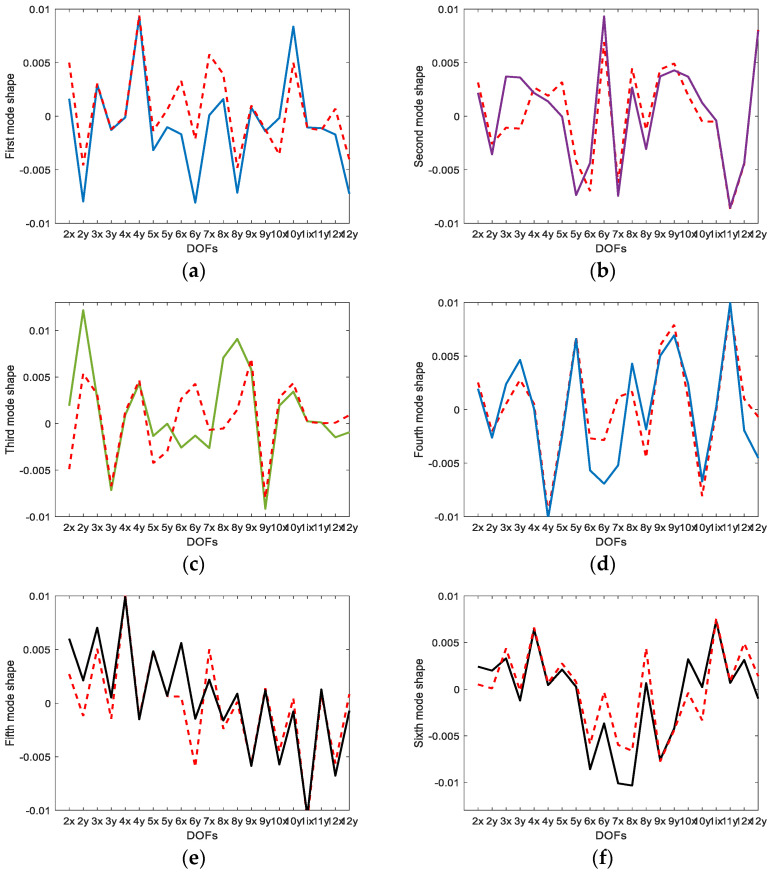
Comparison of estimated and actual mode shapes (strain mode shape-based approach): (**a**) first mode; (**b**) second mode; (**c**) third mode; (**d**) fourth mode; (**e**) fifth mode; (**f**) sixth mode. The solid line indicates the actual mode shape and the dotted line is the estimated mode shape.

**Table 1 sensors-23-02738-t001:** Final sensor positions.

	EI Method	G-EI Method
Final sensor positions	2-x, 4-y, 7-x, 8-x, 9-y, 11-y	2-x, 2-y, 3-y, 4-y, 9-y, 11-y

The number indicates the node number, and the letters x and y indicate the measurement direction.

## Data Availability

The data used to support the findings of this study are included within the article.

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
