# Peer review of "An Optimal Strain Gauge Layout Design for the Measurement of Truss Structures"

_sensors, 2023, doi:10.3390/s23052738_

Round 1

Reviewer 1 Report

The manuscript proposed a method to design the sensor layout with reduced number of sensors; however, I have the following major comments:

1. No validation shown to the proposed method against finite element analyses neither measured responses.

2. The manuscript did not mention what type of damage or defect is intended to be captured by the proposed sensor layout. This shall be included and described in the design of sensors layout.

3. The research shall cite more references about truss bridges behavior because truss bridges are not subjected to axial stresses only.

4. In the Introduction section, first sentence, Any reference for this information?

5. Introduction section, 2nd paragraph, why using high number of sensors is impractical? Please expand. Also, what is the definition of large number of sensors.

6. Introduction section, 3rd paragraph, please provide references.

7. The manuscript mentioned measured and actual displacement? What is meant by these terms?

9. Please add the following references to the manuscript:

Tobias, D. H., Foutch, D. H., & Choros, J. (1993). Investigation of an Open Deck Through-Truss Railway Bridge: Work Train Tests

Yi, T., Li, H., & Gu, M. (2011). Optimal Sensor Placement for Structural Health Monitoring Based on Multiple Optimization Strategies. The Structural Design of Tall and Special Buildings, 20(7), 881-900. doi:10.1002/tal.712

Papadimitriou, C. (2004). Optimal Sensor Placement Methodology for Parametric Identification of Structural Systems. Journal of Sound and Vibration, 278(4), 923- 947.

Meo, M., & Zumpano, G. (2005). On the Optimal Sensor Placement Techniques for a Bridge Structure. Engineering Structures, 27(10), 1488-1497. doi:10.1016/j.engstruct.2005.03.015

DelGrego, M. R., Culmo, M. P., & DeWolf, J. T. (2008). Performance Evaluation through Field Testing of Century-Old Railroad Truss Bridge. Journal of Bridge Engineering, 13(2), 132-138. doi:2(132)

Al-Emrani, M., Akesson, B., & Kliger, R. (2004). Overlooked Secondary Effects in Open-Deck Truss Bridges. Structural Engineering International, 14(4), 307-312

Reviewer 2 Report

The paper uses the algorithm to optimize the arrangement of strain sensors in truss structure. The proposed algorithm can obtain sufficient health information of truss structure with a small number of sensors.

Suggestions:

(1) The redundancy in the introduction part of the manuscript needs to be modified.

(2) The pictures in the manuscript are not very readable and need to be modified, especially Figure 4.

(3) It is mentioned in the paper that the selection of sensors is very important and needs to be explained in detail. It is suggested that different sensors should be selected for experiments to prove the difference of different sensor layout

(4) Academic nouns need to be added to DOF.

Round 2

Reviewer 1 Report

Dear Authours,

For both examples presented, please add more figures shown results. Please add comparisons between mode-shape for all considered methods in addition to mode-shape from any finite element analyses for the truss under investigation. 

Showing differences on mode-shape does not represent the results well. Please add new figures with multiple plot each showing mode-shape of the described methods compared against a plot of mode-shape from finite element analyses.
